# Study on the Effect of Asphalt Static Conditions on the Tensile Properties of Acidic Aggregate Hydraulic Asphalt Concrete

**DOI:** 10.3390/ma17112627

**Published:** 2024-05-29

**Authors:** Lei Bao, Min He, Shu Wang, Xinshuang Wu

**Affiliations:** 1School of Civil Engineering and Architecture, Xi’an University of Technology, Xi’an 710048, China; lbao1013@163.com; 2Power China Northwest Engineering Corporation Limited, Xi’an 710065, China; 13102213129@163.com (S.W.); wuxins@nwh.cn (X.W.); 3State Key Laboratory of Eco-Hydraulics in Northwest Arid Region of China, Xi’an University of Technology, Xi’an 710048, China

**Keywords:** asphalt concrete, bond strength, digital images, numerical simulation, modulus

## Abstract

Hydraulic asphalt concrete is known for its excellent seepage control performance and strong deformation resistance. This engineering material has widespread applications in the seepage control structures of hydraulic buildings. Recent projects have investigated the use of acidic aggregates to improve economic efficiency. However, they have also highlighted the weaker adhesion between acidic aggregates and asphalt, which necessitates stringent construction process control. This study investigates the impact of resting conditions on the tensile properties of acidic aggregate hydraulic asphalt concrete. The results of the tensile testing indicate that the storage time significantly affects the performance of asphalt concrete. The tensile strength of the specimens without anti-stripping agents decreased from 1.711 MPa to 0.914 MPa after resting periods of 0, 10, 20, and 30 days. The specimens treated with anti-stripping agents also showed a decrease in tensile strength over time, similar to the trend observed in the previous specimens. Digital specimen simulations indicated a decrease in cohesion between the asphalt and the aggregate from 5.375 MPa to 2.664 MPa after 30 days, representing a reduction of 50.44%. To counteract the effect of the storage time on the bonding between acidic aggregates and asphalt, this study recommends reducing the grading index and maximum size of aggregates, decreasing the coarse aggregate content, and selecting smooth aggregate shapes.

## 1. Introduction

Hydraulic asphalt concrete primarily consists of aggregates and asphalt and is renowned for its impermeability and robust deformation capacity [1,2,3]. This attribute renders it suitable for diverse impermeable structures, such as retaining walls and impermeable panels in pumped storage power stations [4]. Seepage control structures crafted from hydraulic asphalt concrete offer notable economic efficiency and facilitate an uncomplicated repair process [5,6]. They maintain a prominent position in hydraulic seepage control endeavors, both domestically and internationally.

Numerous investigations have revealed that water and rock dams may encounter adverse consequences when subjected to elevated external loads, such as an excessively thick cover layer or specific site conditions [7,8]. These effects include uneven settlement or deflection of the dam foundation [5,9,10]. Regions such as the counter-arc segment of hydraulic asphalt concrete panels and structural intersections undergo varying degrees of tensile stress, resulting in the formation of cracks [11,12,13]. These cracks can culminate in brittle fracture, diminishing the service life and overall durability [14]. Additionally, as the height of the earth and rock dam increases, the peak tensile stresses and their area of influence also increase [15]. The stability and safety of earth and rock dams substantially diminish when functional damage occurs to a structure under the extreme external loading of hydraulic asphalt concrete [16]. Tensile properties are considered crucial indicators of practical performance in asphalt concrete engineering [17].

The performance of configured asphalt may undergo alterations under different static conditions due to unforeseen factors. Consequently, it is crucial to conduct research on the tensile properties of acidic aggregate hydraulic asphalt concrete under diverse static conditions and other construction control parameters.

Asphalt is a polar substance, and the strength of its polarity and the content of its surface-active substances, such as asphalt acid and asphalt anhydride, play a role in its adhesion to acidic stones. When asphalt and acidic stones come into contact, the asphalt cannot undergo a chemical reaction with the acidic stone. Instead, it produces intermolecular forces, which entails physical adsorption. This results in weak adhesion. When in contact with alkaline stone materials, asphalt can undergo a chemical reaction, resulting in the formation of an insoluble compound. This process is known as chemical adsorption, which is a stronger form of adhesion than physical adsorption. Therefore, it is of particular importance to enhance the understanding of chemical adsorption. Inadequate adhesion between asphalt and aggregates can lead to the detachment of the asphalt from the aggregates, resulting in the structural instability of the asphalt concrete [18]. To address this concern, certain projects have experimented with the incorporation of anti-stripping agents into asphalt, aiming to enhance the adhesion between asphalt and acidic aggregates. The inclusion of anti-stripping agents significantly ameliorates the issue of poor adhesion between asphalt and acidic aggregates, rendering them widely employed in hydraulic asphalt concrete projects where acidic aggregates are extensively utilized. Furthermore, researchers have placed considerable emphasis on the widespread adoption of anti-stripping agents to effectively mitigate the problem of suboptimal adhesion between asphalt and acidic aggregates [19]. The anti-stripping agents enhance the adhesion of the stones to the binder by creating chemical bridges between the two components [20]. The impact of incorporating anti-stripping agents on enhancing the adhesion between asphalt and aggregates has been validated by Rossi through the implementation of contact angle and boiling test measurements [21]. Chen et al. [22] investigated the relatively inferior mechanical properties of asphalt concrete and discovered that this material exhibited heightened vulnerability to temperature fluctuations, particularly in regions characterized by consistently low temperatures during construction. This susceptibility can lead to various issues, including the cracking, deformation, and surface damage of asphalt concrete. Wang et al. [23] conducted compressibility tests to examine the mechanical properties of asphalt concrete under low-temperature conditions and underscored the importance of acknowledging and managing temperature effects to ensure the durability and quality of asphalt concrete. By subjecting it to varying temperatures, Zhang et al. [24] observed that temperature indeed influences the mechanical properties of asphalt concrete. Chen et al. [25] scrutinized asphalt with different oil/stone ratios at varying temperatures and concluded that temperature affects the stress and strain of asphalt concrete specimens. When the temperature plummeted to 0 °C and below, the asphalt concrete exhibited brittle damage and stress-softening behavior. Conversely, when the temperature exceeded 0 °C, the asphalt concrete demonstrated stress-hardening characteristics. Noroozi et al. [26] determined that temperature exerted a significant effect on the dynamic elastic modulus of asphalt concrete, with the modulus decreasing as the temperature increased. Cheng et al. [27] underscored the necessity of the tailored design and construction of asphalt concrete under different temperature conditions, as its performance varies accordingly. In terms of static conditions, both domestic and international scholars have conducted studies on the following aspects. Pais Jorge [28] observed that prolonged storage at 180 °C had detrimental effects on asphalt mixtures. Yang [29] analyzed variations in asphalt performance under different static storage temperatures and durations, including changes in viscosity, three major indices, and elastic recovery.

Guo et al. [30] conducted aging tests and Brinell viscosity assessments to examine the effects of the mixing time and mixing temperature on rubber asphalt’s performance, both before and after aging. Their findings revealed that both the mixing time and the mixing temperature exerted substantial influences on the aging performance of rubber asphalt. Specifically, elevated mixing temperatures accelerated the aging process, resulting in more pronounced performance deterioration. Similarly, an increased mixing time also accelerated rubber asphalt aging, leading to greater degradation [31]. Furthermore, the asphalt viscosity exhibited notable fluctuations in response to temperature variations and different mixing times. Michon [32] determined that long-term low-temperature storage could impact asphalt due to temperature variations.

Predicting the mechanical properties of concrete has perpetually intrigued researchers. Currently, most fracture studies on asphalt concrete involve experimental research. However, several scholars have employed numerical simulation methods to predict the mechanical properties of asphalt concrete. Taesun [33] proposed a multiscale asphalt pavement analysis approach employing a multiscale thermomechanical finite element model to connect the heterogeneous local scale with the global homogeneous asphalt pavement scale. The use of discrete damage models, such as the cohesive interface cell model, is prevalent for simulating solid fracture damage evolution. This model can be readily implemented in computer programs. Moreover, diffuse damage models obviate the need for explicit crack path-tracking algorithms and can model multiple cracks and complex crack types. Baziar [34] employed numerical modeling and material parameters derived from laboratory tests to scrutinize the seismic behavior of the Miguelan Dam, which is characterized by an asphalt concrete core. The primary objective of this study was to develop discrete damage models, including the cohesive interface unit model, to simulate solid fracture and explore the fracture behavior of asphalt concrete.

In the domain of research pertaining to tensile properties, there has been a conspicuous emphasis on alkaline aggregates, while investigations into acidic aggregates have been relatively scant, resulting in a dearth of data. Furthermore, research into the tensile performance of acidic aggregates under various static conditions in asphalt remains limited. Additionally, prevailing methods for testing tensile performance primarily rely on physical experiments, posing challenges in terms of comprehending the underlying factors influencing tensile performance. The findings of this study offer a valuable predictive and regulatory framework for materials employed in acid aggregate hydraulic projects, enabling them to effectively navigate uncertainty and uncontrollable circumstances.

To address these concerns, this study aimed to investigate the tensile properties of hydraulic asphalt concrete containing acidic aggregates under diverse static conditions. By combining indoor experiments with numerical simulations, alterations in the tensile properties of asphalt concrete and the formulated regulatory measures were investigated. The principal objective of this investigation was to elucidate the governing principles dictating changes in the tensile properties and propose corresponding regulatory measures.

(1)Research was conducted to assess the static conditions affecting the tensile properties of asphalt concrete. Tensile tests were conducted under different static conditions, including 0 days, 10 days, 20 days, and 30 days, with the objective of analyzing the alterations in the tensile properties of the asphalt concrete under these varying static conditions.(2)To explore the influence of adhesion between aggregates and asphalt on tensile strength and suggest enhancement strategies, numerical simulations were performed using ABAQUS 2023. This study focused on scrutinizing the microlevel influencing factors, including coarse aggregate characteristics such as the aggregate gradation, coarse aggregate content, maximum particle size, and angular index. The tensile strength variations were compared under different influencing factors.

## 2. Materials and Methods

This experiment comprises two distinct segments. In the initial segment, tensile assessments were carried out on asphalt concrete samples following resting periods of 0, 10, 20, and 30 days to determine their tensile strength and compute pertinent parameters.

The subsequent section aims to illustrate the viability of numerical simulation through the amalgamation of physical testing and numerical modeling. To mitigate the influence of temperature and anti-stripping agents, the initial temperature was fixed at 160 °C, and the anti-stripping agent dosage was maintained at 0.6%. The tensile properties of the asphalt concrete were investigated at different time intervals (0 days, 10 days, 20 days, and 30 days) via tensile testing. A comparative analysis was conducted to assess the variations in the tensile strength and modulus.

### 2.1. Materials

The tensile properties of the asphalt concrete were assessed following the procedures outlined in previous tests [10]. The coarse aggregate utilized was artificial aggregate produced from gneiss sourced from Yimeng, Shandong, China. The fine aggregates consisted of artificial sand, while the filler was derived from the screening of artificial sand. Karamay 70 asphalt was employed as the asphalt component.

#### 2.1.1. Asphalt

Among the numerous materials employed in the construction of water projects, Klamath 70 asphalt is renowned for its relatively low cost, which has rendered it a primary consideration in many instances [35,36]. The asphalt model used was Klamath 70 Grade A asphalt, and the parameters of the asphalt are detailed in Table 1.

#### 2.1.2. Coarse Aggregate

Coarse aggregates with particle sizes ranging from 19 to 2.36 mm were carefully chosen for this study. Upon scrutinizing the test results, it became apparent that the selected coarse aggregate demonstrated commendable attributes. Notably, it exhibited robust adhesion to asphalt and displayed a minimal water adsorption rate. These attributes play a pivotal role in maintaining the structural integrity and overall performance of asphalt concrete containing acidic aggregates. Moreover, when evaluating each test parameter individually, it became evident that the coarse aggregate met the stipulated criteria. The detailed parameters of the coarse aggregates are presented in Table 2 for reference.

#### 2.1.3. Fine Aggregates

Fine aggregates (as detailed in Table 3) within the particle size range of 2.36 to 0.075 mm were thoughtfully selected based on their hardness, freshness, and purity. This choice aimed to mitigate any potential detrimental effects on the mechanical properties of the asphalt concrete.

The chosen fine aggregates exhibited a desirable texture and maintained their structural integrity without cracking or decomposition under the influence of heat. Additionally, these aggregates displayed favorable performance characteristics, with the durability and apparent density conforming to the specified requirements.

#### 2.1.4. Fillers

The selection of asphalt concrete fillers should take into account factors such as the fineness, apparent density, and other relevant specifications. The appropriate choice of filler has the potential to optimize the performance and longevity of asphalt concrete. In this particular study, an acidic filler was employed, and the pertinent filler parameters are described in Table 4.

It is evident that the comprehensive assessment of filler properties, including the fineness, apparent density, and other relevant parameters, serves as a reliable measure of quality. These properties fully align with the specified technical requirements, ensuring the reliability and suitability of the filler for the intended application.

#### 2.1.5. Anti-Stripping Agent

The anti-stripping agent (Shui Ke, Chinese Academy of Water Sciences (CAS), Beijing, China) was purposefully formulated to cater to the utilization of acidic aggregates in hydraulic asphalt concrete construction.

The primary constituents of this agent included rosin and various surfactants. These components were carefully blended to align with the demands of water conservancy projects involving hydraulic asphalt concrete. Employing molecular structure design, a novel category of asphalt concrete anti-stripping agent was synthesized. This product notably enhances the adhesion between asphalt and acidic aggregates while minimizing its influence on other asphalt indicators.

The product specification delineates the following characteristics:(1)Good product performance: For ordinary granite, gneiss and other acidic aggregates, a mixed solution of anti-stripping agent and asphalt with a weight ratio of 0.2% to 0.4% has been found to significantly enhance adhesion. In order to meet the hydraulic asphalt concrete construction requirements, the anti-stripping agent parameter needs to be increased to 0.6% of the weight of asphalt to 1.2%. This is necessary in order to ensure the stability of the asphalt and acidic aggregate water.(2)Wide range of adaptation: SK-A asphalt concrete anti-stripping agent is not only suitable for hydraulic asphalt concrete construction but can also be applied to road asphalt concrete construction.(3)Good applicability: Solid particles at room temperature, easy to transport. Melting point of 60 °C, easily soluble in asphalt, easy to use. The boiling point is 230 °C, and the high-temperature performance and stability are excellent. The 130 °C asphalt can be sealed for storage for up to 30 days without any significant loss of adhesion.(4)Environmentally friendly: The product contains no volatile components and no pollution, and the construction process has a minimal impact on human health.

#### 2.1.6. Grading

According to Ding Pu Rong’s formula [38], the asphalt/concrete mix ratio necessitates the consideration of several key factors, including the gradation index, filler quantity, and oil/rock ratio. The selection of an appropriate asphalt/concrete mixing ratio should be contingent upon project-specific requirements and material characteristics. A comprehensive assessment of the aforementioned factors is crucial for adjustment and final determination.

Several factors that influence the performance of asphalt concrete are worth noting:(1)Properties of the raw materials, including asphalt, coarse aggregate, fine aggregate, and filler;(2)Aggregate grading;(3)Filler quantity, denoting the proportion of filler within the mineral composition;(4)The oil/stone ratio is defined as the ratio of the asphalt mass to the aggregate mass in the asphalt mixture.

The parameters related to the aggregate grading include the maximum aggregate particle size, grading index, coarse and fine aggregate proportions, and filler quantity. These can be determined through the maximum dense-grading theory and the formula recommended by the Design Code for Asphalt Concrete Panels and Core Walls for Earth and Stone Dams (DL/T 5411-2009), as calculated below:(1)$$P_{i}=F+(100-F)\frac{d_{i}^{r}-d_{0.075}^{r}}{d_{\max}^{r}-d_{0.075}^{r}}$$
where:*P*_i_—passage rate of sieve *d_i_*, %;*F*—amount of filler with a particle size of less than 0.075 mm, %;*d*_max_—maximum particle size of the mineral, mm;*d*_i_—a certain sieve size, mm;*d*_0.075_—maximum particle size of the filler, 0.075 mm;*r*—grading index.

The configuration of this test was grounded in prior engineering expertise and the results of experimental investigations. The chosen mixture ratio consisted of a 7% oil-to-gravel ratio, a 9% filler dosage, and a grading index of 0.5. The detailed parameters are outlined in Appendix A.

In accordance with the validation test criteria, the tensile testing will predominantly focus on different resting durations.

### 2.2. Method

#### 2.2.1. Equipment and Conditions

(1)Specimen preparation and specimen size: The asphalt mixture was processed into sizeable plate specimens measuring 12 cm × 6 cm × 25 cm. For each layer of the specimen, 105 impacts were performed using a Marshall standard hammer. Subsequently, after the plate specimens were naturally cooled, they were cut into 4 cm × 4 cm × 20 cm tensile specimens to determine their density and calculate their porosity. These cut specimens were then affixed to the tensile specimen chuck using high-strength adhesive, allowed to stabilize for 24 h, and subjected to temperature equilibration before testing.(2)Testing temperature: Maintained at 5 ± 0.5 °C.(3)Testing loading rate: Set at 1.0 mm/min.

The tensile strength and tensile strain were computed utilizing the following formulas:(2)$$R_{t}=\frac{P}{A}$$
(3)$$\varepsilon_{t}=\frac{\delta_{t}}{L}$$
where:*R_t_*—tensile strength, in MPa;*ε_t_*—strain at maximum stress of the specimen, %;*δ_t_*—axial tensile deformation, mm;*P*—specimen subjected to the axial maximum load, N;*A*—specimen section area, mm^2^;*L*—specimen axial side marking distance, mm.

The tensile deformation modulus was calculated by the following formula:(4)$$E_{t}=\frac{R_{t}}{\varepsilon_{t}}$$
where:*E*_t_—tensile strength, MPa;*R*_t_—a certain tensile stress, MPa;*ε_t_*—strain at maximum stress of the specimen, %.

Calculation of the tensile deformation modulus involved determining the slope of the section within the load–deformation curve, specifically within the range of 0.1 to 0.7 of the maximum load *R*_t_, in cases where the curve was not linear.

#### 2.2.2. Ontological Relationships

Cohesive zone modeling (CZM) was utilized in the modeling to replicate microscale damage. Central to the CZM was the law of traction separation, which established the connection between the applied traction force and the relative displacement, specifically the crack mouth opening displacement (CMOD), between two surfaces. Figure 1 illustrates the bilinear traction separation law, outlining the traction force as a function of the CMOD. When the initial damage threshold was met (as depicted in Figure 1), the element’s fracture energy was released, triggering crack initiation. Subsequently, the element forfeits its load-bearing capacity, transferring the load to adjacent elements and resulting in crack propagation. In this study, three different traction separation laws were employed: one for the bending mode and two for the shear mode. However, given the two-dimensional nature of the developed model, only two failure modes were considered: one for bending and the other for shear. For the sake of simplification, it was assumed that these modes were identical, thus adopting a bilinear traction separation law.

Subsequently, by leveraging the derived tensile deformation modulus and tensile strength, parameter inversion was conducted in ABAQUS to determine the bond strength of the asphalt mortar–coarse aggregate interface under various modulus and tensile strength conditions.

### 2.3. Study of the 2D Model Interface Parameters

#### 2.3.1. Interfacial Parameter Studies

From a microscopic perspective, crack propagation within asphalt concrete could traverse through the coarse aggregate, the asphalt mortar, and the asphalt mortar–coarse aggregate interfaces shown in Figure 2a. However, it is important to note that the strength of the coarse aggregate significantly exceeded that of the asphalt mortar and the asphalt mortar–asphalt mortar interfaces shown in Figure 2b. Consequently, the coarse aggregate typically remained intact and did not undergo fracture.

#### 2.3.2. Asphalt Mortar–Coarse Aggregate Interface Research

The strength of the asphalt mortar–coarse aggregate interface was notably inferior to that of the asphalt mortar–asphalt mortar interface. Consequently, it constituted the more vulnerable segment within the asphalt concrete specimens. In other words, the asphalt mortar–coarse aggregate interface was more susceptible to cracking under external loading. The investigation in this subsection focused on regulating and examining both the fracture energy and the bond strength of the asphalt mortar–coarse aggregate interface. These two parameters shown in Table 5 influence the tensile properties of asphalt concrete specimens, with a particular focus on how different aspects of the asphalt mortar–coarse aggregate interface impact these properties.

Initially, while maintaining a constant bond strength of 0.030 N·m at the asphalt mortar–coarse aggregate interface, values of 0.045 N·m, 0.060 N·m, and 0.075 N·m were successively chosen for the fracture energy parameter. Subsequently, with the fracture energy parameter of the asphalt mortar–coarse aggregate interface held constant, values of 1.4 MPa, 1.9 MPa, 2.4 MPa, and 2.9 MPa were selected for the bond strength in sequence. Employing the numerical testing method outlined in the preceding subsection and utilizing ABAQUS for the tensile testing, alterations in the peak stress subsequent to numerical simulation of the asphalt concrete specimen’s tensile test and the energy expended for the specimen’s failure under varying fracture energy and bond strength conditions at the asphalt mortar–coarse aggregate interface were determined. To facilitate a more comprehensive examination of the influence of different fracture energies and bond strengths on the fracture performance of asphalt concrete specimens at the asphalt mortar–coarse aggregate interface, a wide range of fluctuating values was considered, aligning with specific engineering scenarios for in-depth analysis.

As the bond strength increased during the tensile test, the peak stress gradually increased. The selected bond strength values for this simulation ranged from 1.4 MPa to 1.9 MPa, 2.4 MPa, and 2.9 MPa, corresponding to peak stress values of 1.097 MPa, 1.268 MPa, 1.439 MPa, and 1.610 MPa, respectively. This observation underscores the pivotal role of bond strength as a key factor influencing the peak stress value. Conversely, when the fracture energy parameter was increased from 0.030 N·m to 0.075 N·m, the corresponding peak stress values in the four cases were 1.264 MPa, 1.268 MPa, 1.275 MPa, and 1.281 MPa (Figure 3). Notably, the peak stress values associated with the four fracture energy values exhibited minimal disparities. However, the effect on the softened section was significant. With an increase in the fracture energy, the rate of stress reduction within the softened section became progressively slower. This phenomenon can be attributed to the fact that, in the context of tensile strength determination, the fracture energy parameter does not directly affect crack opening, nor does it influence the elastic modulus or tensile strength of the specimen. Rather, the alteration in the fracture energy primarily affects the softening section of the concrete. When the fracture energy is higher, the unit requires more energy for cracking, resulting in increased destructive displacement within the unit and consequently leading to a slower rate of decline in stress within the softened section.

#### 2.3.3. Asphalt Mortar–Asphalt Mortar Interface Research

The asphalt mortar–asphalt mortar interface exhibited a propensity for crack formation within the asphalt concrete specimens. To investigate the influence of the asphalt mortar–asphalt mortar interface on the mechanical properties of asphalt concrete specimens under varying fracture energies and bond strengths, different fracture energies and bond strengths were applied to this interface. This enabled an assessment of the effect of the asphalt mortar–asphalt mortar interface on the mechanical properties under these varied conditions.

Following a similar numerical simulation approach as before, with the bond strength of the asphalt mortar–asphalt mortar interface held constant, values of 0.040 N·m, 0.090 N·m, 0.140 N·m, and 0.190 N·m were sequentially selected for the fracture energy parameter. Subsequently, with the fracture energy parameters of the asphalt mortar–coarse aggregate interface unchanged, values of 3.3 MPa, 3.8 MPa, 4.3 MPa, and 4.8 MPa were chosen for the bond strength. This enabled an examination of the effects of the fracture energy and bond strength on the fracture properties of asphalt concrete specimens under different parameters at the asphalt mortar–asphalt mortar interface. Additionally, the simulation results of the stress-strain curves from the tensile tests were analyzed. Like for the parameters for the asphalt mortar–coarse aggregate interface, a wide range of fluctuating parameter values was considered for the asphalt mortar–asphalt mortar interface to comprehensively assess the influence of different fracture energies and bond strength values on the fracture properties of the asphalt concrete specimens. This approach was employed to provide a more intuitive and distinct understanding of the parameter effects through numerical testing.

The interfacial strength of the asphalt mortar–asphalt mortar interface increased from 3.3 MPa to 3.8 MPa, 4.3 MPa, and 4.8 MPa (Figure 4). Correspondingly, the peak stress associated with this interfacial strength exhibited an increase from 1.136 MPa to 1.238 MPa, 1.331 MPa, and 1.524 MPa. In contrast, as the fracture energy increased from 0.075 N·m to 0.09 N·m, 0.105 N·m, and 0.190 N·m, the peak stress corresponding to these values demonstrated minimal variation. During the softening stage, the descending trend gradually levelled off.

Through a two-dimensional numerical simulation of the asphalt concrete tensile behavior considering different interfacial parameters for the fracture energy and bond strength, it became evident that the asphalt mortar–coarse aggregate interface exerted a more substantial influence on the mechanical properties of asphalt concrete. The tensile strength was predominantly determined by the interfacial strength of the asphalt mortar–coarse aggregate interface.

### 2.4. Numerical Simulation

#### 2.4.1. Comparison between 2D Models and Actual Working Conditions

Following the ABAQUS-based numerical simulation of the two-dimensional asphalt concrete model tensile test, it was necessary to undertake a comparative analysis between the numerical simulation results and the results obtained from physical tests. This comparative assessment served as a means to validate the feasibility and accuracy of the numerical simulation test.

The energy consumption observed in the actual experimental results was 193.6 mJ (Figure 5), while the numerical simulation yielded a value of 169.9 mJ, resulting in a 12% difference between the two test methods. Notably, the peak stress values in the actual test results and the numerical simulation curves closely aligned. However, it is worth mentioning that the energy consumption recorded in the physical test exceeded that of the numerical simulation test. This discrepancy can be attributed to two factors.

First, it is possible that the modulus used for reference in the bond strength inversion was greater than the actual test’s modulus, contributing to the variation in the results. Second, the span diameter in the actual test was approximately 160 mm, while the numerical simulation of the tensile specimen featured a larger span diameter of 220 mm. This difference in the span diameter may have also contributed to variations in the strain values between the two methods.

#### 2.4.2. Inversion of Interface Parameters

The bond strength, derived through the inversion of the tensile modulus shown in Table 6, exhibited a progressive decrease in the interfacial strength of the asphalt mortar–coarse aggregate interface as the storage time increased. This observation further confirmed that the mechanical properties of the asphalt concrete specimens decreased with a prolonged storage time. By utilizing a fitting approach, it is possible to extrapolate the bond strength for any day within the range of 0 to 30 days, facilitating the calculation of the tensile strength based on these extrapolated values. Detailed changes in the relationship are shown in Figure 6.

## 3. Results and Discussion

### 3.1. Tensile Test Results

From Figure 7, it can be found that the tensile strength of the specimens treated with the anti-stripping agent surpassed that of the group without it. Furthermore, the density of the specimens exhibited a decreasing trend as the storage time increased, indicating an increase in porosity. In other words, Table 7 shows that the adhesion between the asphalt and the acidic aggregate weakened with a prolonged storage time. The overall tensile strength of the asphalt concrete tensile specimens decreased, while the tensile modulus gradually increased. This observation led to the conclusion that the adhesion between the asphalt and acidic aggregate decreased as the storage time increased.

### 3.2. Research on the Modulation of the Tensile Strength of Hydraulic Asphalt Concrete with an Acidic Aggregate

The investigation of the interface parameters in the 2D fine-scale model revealed that the strength of the asphalt concrete was predominantly influenced by the parameters at the asphalt mortar–coarse aggregate interface. The effect of the asphalt mortar–coarse aggregate interface clearly varied depending on the coarse aggregate characteristics (such as the coarse aggregate gradation, content, maximum particle size, and angular index) and other factors.

Consequently, a numerical simulation was conducted using ABAQUS to explore the influence of various factors, including the coarse aggregate properties, on the tensile properties of asphalt concrete. Building upon these findings, corresponding measures for controlling the tensile strength were proposed.

#### 3.2.1. Effect of Coarse Aggregate Gradation on Tensile Strength

The gradation of coarse aggregates plays a significant role among the factors influencing asphalt concrete performance. The coarse aggregate content plays a pivotal role in determining the mechanical properties and service life of asphalt concrete. For this study, the No. 1 mixture was used as a reference, and modifications were made to the coarse aggregate gradation based on Ding Pu Rong’s formula, aligning with the specifications outlined in the DL/T5326-2018 [39]. The aggregate gradation for five different scenarios was obtained, and the asphalt concrete grading for the converted two-dimensional digital model is presented in Appendix A.

As the gradation increased from 0.3 to 0.7, the corresponding peak stress decreased from 1.431 MPa to 1.145 MPa (Figure 8). This phenomenon can be attributed to the fact that as the gradation index increased, the proportion of larger-sized aggregates also increased. Consequently, with an increasing coarse aggregate particle size, the perimeter of the unit area decreased. As a result, the crack paths generated during cracking were shorter, leading to lower energy consumption. This underscores the importance of considering the influence of the gradation size on the mechanical properties of asphalt concrete during grading design.

#### 3.2.2. Effect of Coarse Aggregate Content on Tensile Strength

In continuation of the No. 1 mix ratio, variations were introduced to the coarse aggregate content for this subsection. The selected coarse aggregate contents included 20%, 30%, 40%, and 50%, leading to the creation of four different numerical asphalt concrete specimens, as illustrated in Appendix A. It is essential to note that the coarse aggregate content used in this study does not align with the typical range found in actual projects. Instead, these values were chosen to investigate the effect of the coarse aggregate content on the tensile strength of asphalt concrete over a wider range of fluctuations. Following the previously employed methodology, tensile test simulations were conducted to obtain the stress-strain curves for asphalt concrete specimens with varying coarse aggregate contents. Detailed results of the test are shown in Figure 9.

As the coarse aggregate content increased, the peak stress values in the asphalt concrete specimens exhibited varying degrees of decrease, while the elastic modulus gradually increased. This phenomenon can be attributed to the following factors:

Increase in the Coarse Aggregate Content: As the content of coarse aggregate increased, the fracture paths along the asphalt mortar–asphalt mortar interface gradually reduced the proportion of the asphalt mortar–coarse aggregate interface. In comparison, the asphalt mortar–asphalt mortar interface was more susceptible to cracking, representing a more fragile component. Consequently, this shift inevitably led to a decrease in the stress peak.

Optimal Coarse Aggregate Content: However, an excessively low coarse aggregate content can hinder the formation of a robust asphalt concrete structure. Therefore, in asphalt concrete design, it is essential to strike a balance between achieving a good skeletal structure and optimizing the mechanical properties of asphalt concrete. Adjusting the coarse aggregate content appropriately can help achieve this balance.

#### 3.2.3. Effect of the Maximum Coarse Aggregate Size on the Tensile Strength

While maintaining a constant asphalt–concrete grading index, the maximum particle size of the coarse aggregate in the asphalt–concrete mixture was adjusted. The coarse aggregate sieve rates for each grade were calculated based on Ding Pu Rong’s method. The selected maximum particle sizes for the coarse aggregate were 19 mm, 16 mm, 13.2 mm, 9.5 mm, and 4.75 mm, and the corresponding calculated sieve rates for the different maximum particle sizes are presented in Appendix A.

The findings indicate a gradual decrease in the peak stress of the asphalt concrete specimens with an increase in the maximum coarse aggregate size (Figure 10). This observation further confirmed that the previous gradation was too large and did not contribute to improving the tensile strength performance. The reason for this phenomenon may be that as the maximum coarse aggregate size increases, the proportion of large-grained coarse aggregate also increases. Larger coarse aggregate sizes lead to more asphalt mortar–coarse aggregate interfaces, which are fragile interfaces in asphalt concrete. Consequently, the tensile stress required to cause damage to the tensile specimen under external loads decreases.

This suggests that increasing the content of coarse aggregates within the range of 4.75 mm to 2.36 mm can enhance the tensile strength of asphalt concrete while avoiding issues related to difficult compaction and excessive porosity associated with larger coarse aggregate particle sizes. Notably, there are certain correlations among the influencing factors, such as the coarse aggregate gradation, coarse aggregate content, and maximum coarse aggregate size. Consequently, the conclusions drawn regarding the influence of these factors on the mechanical properties of asphalt concrete are in alignment.

#### 3.2.4. Effect of the Coarse Aggregate Prismatic Index on the Tensile Strength

The significance of the aggregate shape in industrial production has been understudied, particularly concerning the various shapes of aggregates commonly used in actual production. Therefore, it is crucial to comprehend how different aggregate shapes influence the mechanical properties of asphalt concrete [40,41]. While controlling the shape of coarse aggregates during actual tests may be challenging, numerical simulation offers an effective means of controlling the effect of the aggregate shape on the mechanical properties of asphalt concrete. This approach is essential for optimizing asphalt/concrete mix ratios and enhancing the mechanical performance of asphalt. Furthermore, the influence of the coarse aggregate shape on the mechanical properties of asphalt concrete should not rely solely on the strength grade; it is also be related to the specific environmental and usage conditions. Hence, a diverse array of factors must be considered to accurately assess the effect of the coarse aggregate shape on asphalt concrete.

Studies have consistently demonstrated that the coarse aggregate angularity index is one of the most critical factors affecting asphalt concrete performance. The coarse aggregate angularity index quantifies the sharpness of coarse aggregate particle shapes and the number of angles they possess. High angular index coarse aggregate particles have the following effects:Increased Friction: Coarse aggregate particles with a high angular index exhibit larger contact areas with cementitious materials (e.g., asphalt) in the concrete, leading to enhanced bonding and friction, which contributes to the tensile strength of the asphalt concrete.Enhanced Cohesion: High angular index coarse aggregate particles can securely lock cementitious material, providing better cohesion and thus improving the tensile properties of asphalt concrete.Impact on Void Ratio: Coarse aggregate particles with a high angular index efficiently fill voids in the concrete, reducing the void ratio, enhancing the compactness, and strengthening the asphalt concrete.

In this study, the angularity index served as the evaluation criterion for quantifying the coarse aggregate angularity and was calculated using the following formula. A round coarse aggregate would yield an angularity index of 1, whereas irregularly shaped coarse aggregates would result in an angularity index greater than 1. The greater the angularity index is, the more angular the shape of the aggregate.
(5)$$AI=\frac{C^{2}}{4\pi S}$$
where:*C*—perimeter of coarse aggregate*S*—coarse aggregate area

The calculation of the prismatic indices for various geometric shapes, including ortho-quadrilaterals, ortho-hexagons, ortho-octagons, ortho-decagons, and circles, was performed in conjunction with Equation (5). The results of these calculations are presented in Appendix A.

Additionally, the number of prongs on coarse aggregates of different particle sizes represents a crucial microstructural aspect of coarse aggregates and is related to the number of asphalt mortar–coarse aggregate interfaces. Since this study utilized a two-dimensional model for asphalt concrete and assumed coarse aggregates to be irregular polygons, an investigation into the number of prongs with different quantities was conducted.

Starting with the circular coarse aggregate digital specimens, various geometric shapes, including orthogonal, ortho-hexagonal, ortho-octagonal, and ortho-decagonal coarse aggregates, were used to generate digital specimens with different prismatic indices. Furthermore, two-dimensional digital specimens were created by altering the grain size under different coarse aggregate prismatic indices, all of which were based on the No. 1 mix ratio.

It is evident that the peak stress of the asphalt concrete specimen gradually increases with the number of prongs (Figure 11). The portion with the least number of prongs exhibited the most significant increase, with its corresponding stress increasing from 1.171 MPa to 1.365 MPa. This phenomenon may be attributed to the fact that coarse aggregates with more prongs possess a larger perimeter per unit area. Consequently, greater energy consumption and external loads are required to induce surface cracking and damage to coarse aggregates. Therefore, as the angular index of coarse aggregate increases, the mechanical properties of asphalt concrete also improve. As the number of external edges of the aggregate decreases, the peak stress values in the tensile test become progressively larger.

This observation may be attributed to two factors. First, for a given aggregate content and two-dimensional area, a decrease in the number of coarse aggregate edges results in a longer perimeter for each coarse aggregate, creating more interfacial cells in the specimen. This, in turn, leads to a decrease in the tensile strength of the specimen. Second, the stress concentration is more pronounced at the corners of polygonal coarse aggregates than at the corners of elliptical and circular coarse aggregates. Among these, the stress concentration in the elliptical aggregates is the second highest. Consequently, the peak stress of the circular aggregate specimens surpasses that of the elliptical aggregate specimens, and the peak stress of the elliptical aggregate specimens exceeds that of the polygonal aggregate specimens.

## 4. Conclusions

In the investigation of the tensile properties of acidic aggregate hydraulic asphalt concrete under different static conditions, both indoor experiments and numerical simulations were conducted. This research focused on the changes in the tensile properties of asphalt concrete and the measures taken to regulate them. The key findings are as follows:(1)Examination of the effect of the storage time on the mechanical properties revealed the following results: After 10, 20, and 30 days of resting, the tensile specimens not adulterated with the anti-stripping agent exhibited a decrease in tensile strength, from 1.711 MPa to 1.573 MPa, 1.073 MPa, and 0.914 MPa, respectively. In contrast, the tensile modulus increased from 83.24 MPa to 116.91 MPa. The tensile strength of the specimens containing the anti-stripping agent decreased from 1.723 MPa to 1.609 MPa, 1.324 MPa, and 1.294 MPa after 10, 20, and 30 days of rest, respectively, while the tensile modulus increased from 85.39 MPa to 113.79 MPa, 119.19 MPa, and 146.46 MPa, respectively. These results suggest that the addition of anti-stripping agents did not alter the decreasing trend in the tensile properties of asphalt concrete with a prolonged storage time.(2)Comparative analysis between the numerical simulation and physical tests revealed that the decrease in the tensile properties could be attributed to adjustments in the interface parameters of the asphalt mortar–coarse aggregate. The cohesion between the asphalt and aggregate decreased significantly from 5.375 MPa to 2.664 MPa over 30 days of resting, representing a 50% reduction. Additionally, the grading indices of the coarse aggregate and coarse aggregate contents and the maximum coarse aggregate particle size were found to influence the peak stress of asphalt concrete. An increase in the number of vertices of coarse aggregate led to a gradual decrease in the peak stress. In conclusion, reducing the grading index and maximum particle size of aggregates, decreasing the content of coarse aggregates, and favoring round and smooth aggregate shapes while reducing porosity were identified as measures to mitigate the adverse effects of the storage time on the adhesion of acidic aggregates to asphalt.

The research carried out in this work on specific ratios for specific aggregates needs further research on different ratios for aggregates with different pH levels. In addition, more advanced equipment was used to directly test the adhesion force.

## Figures and Tables

**Figure 1 materials-17-02627-f001:**
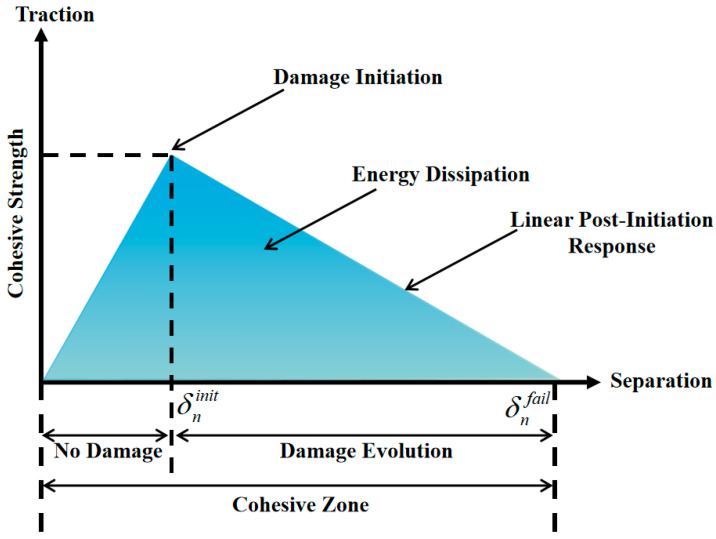
Bilinear traction separation law.

**Figure 2 materials-17-02627-f002:**
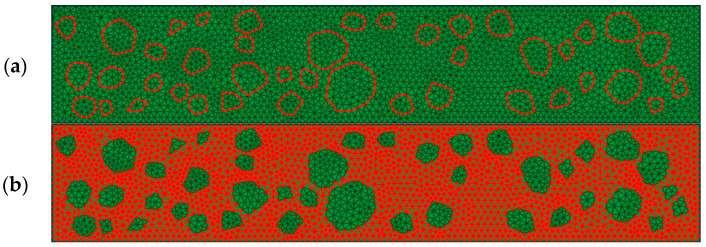
Asphalt concrete tensile digital specimens(The red part is the interface, the green part is the digital specimen): (**a**) Asphalt mortar–coarse interface, and (**b**) asphalt mortar–asphalt mortar interface.

**Figure 3 materials-17-02627-f003:**
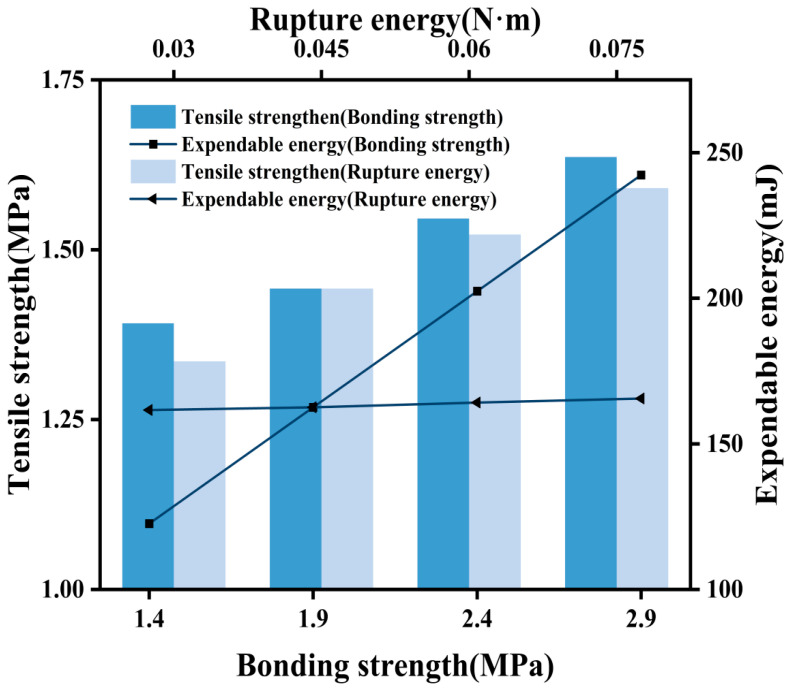
Asphalt mortar–coarse aggregate interface.

**Figure 4 materials-17-02627-f004:**
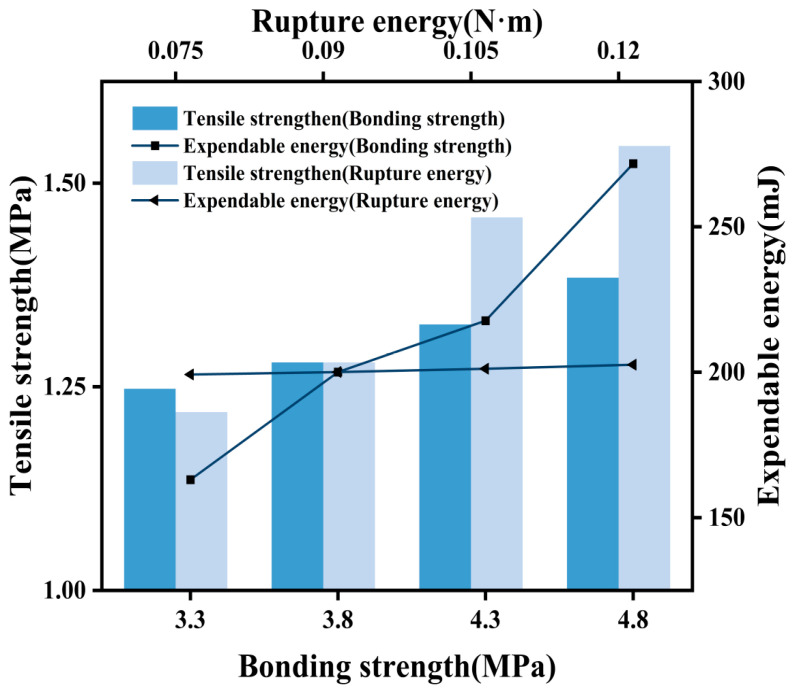
Asphalt mortar–asphalt mortar interface.

**Figure 5 materials-17-02627-f005:**
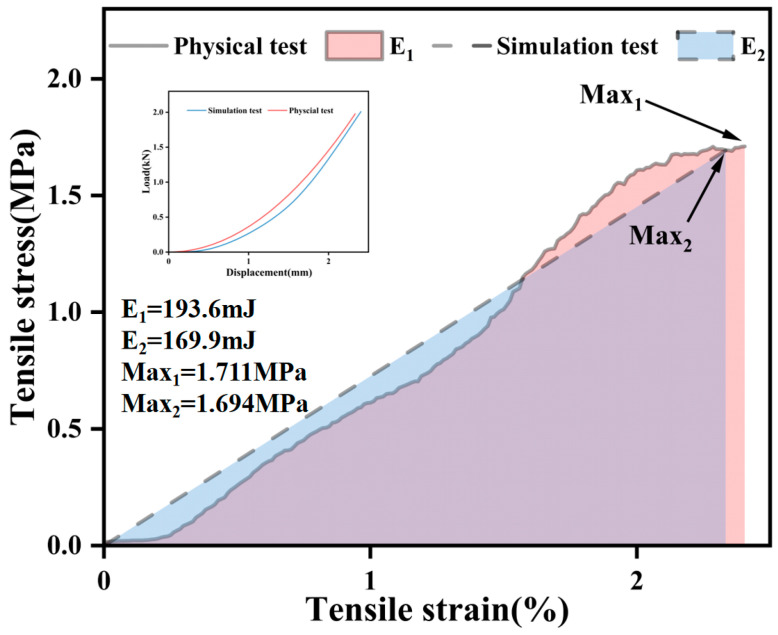
Numerical simulation vs. physical test.

**Figure 6 materials-17-02627-f006:**
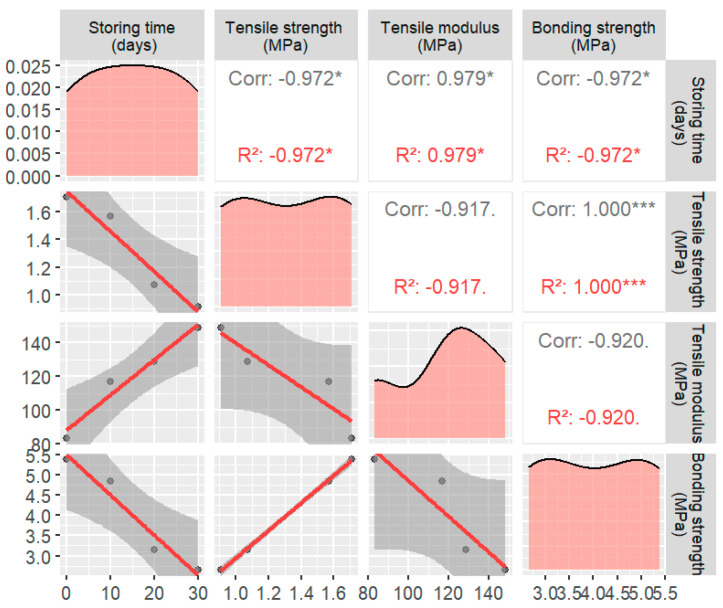
Comparison of asphalt concrete parameters after different static durations (* usually indicates that the *p*-value lies between 0.05 and 0.01, indicating statistical significance, but not very high significance. *** indicates that the *p*-value is less than 0.001, indicating a very high level of statistical significance).

**Figure 7 materials-17-02627-f007:**
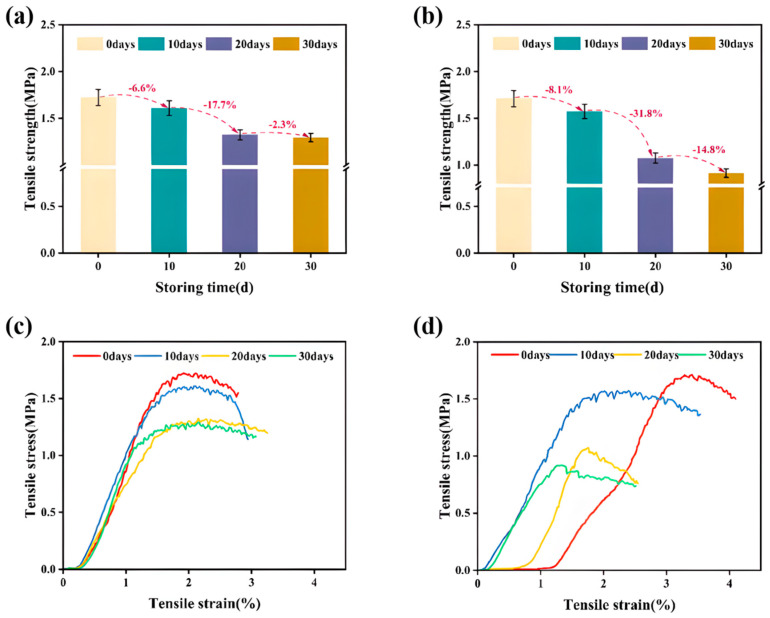
Tensile test data: (**a**) change in tensile strength (with anti-stripping agent), (**b**) change in tensile strength (without anti-flaking agent), (**c**) stress-strain curve (with anti-stripping agent), and (**d**) stress-strain curve (without anti-stripping agent).

**Figure 8 materials-17-02627-f008:**
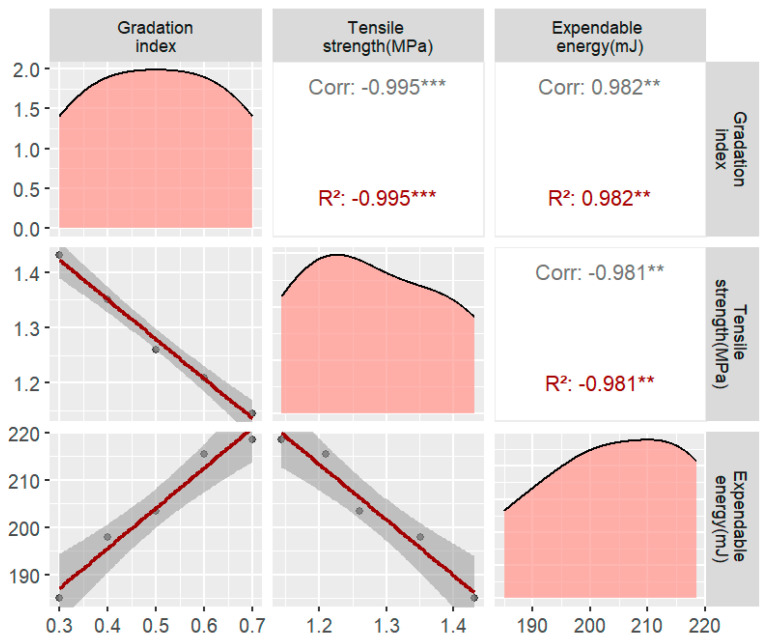
Plot of the tensile strength and energy dissipation for different gradation indices (** probability value of p less than 0.01 indicates high statistical significance. *** indicates that the *p*-value is less than 0.001, indicating a very high level of statistical significance).

**Figure 9 materials-17-02627-f009:**
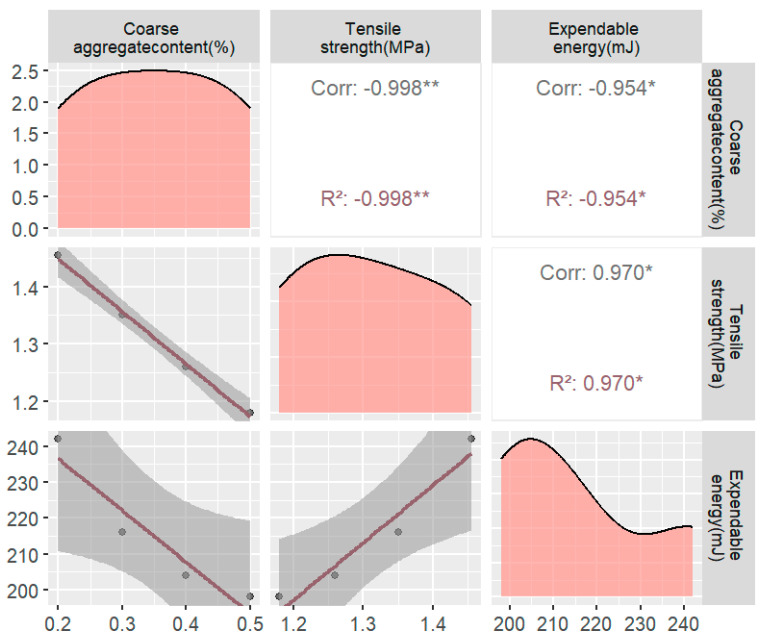
Plot of the tensile strength and energy consumption for different coarse aggregate contents (* usually indicates that the *p*-value lies between 0.05 and 0.01, indicating statistical significance, but not very high significance. ** probability value of *p* less than 0.01 indicates high statistical significance).

**Figure 10 materials-17-02627-f010:**
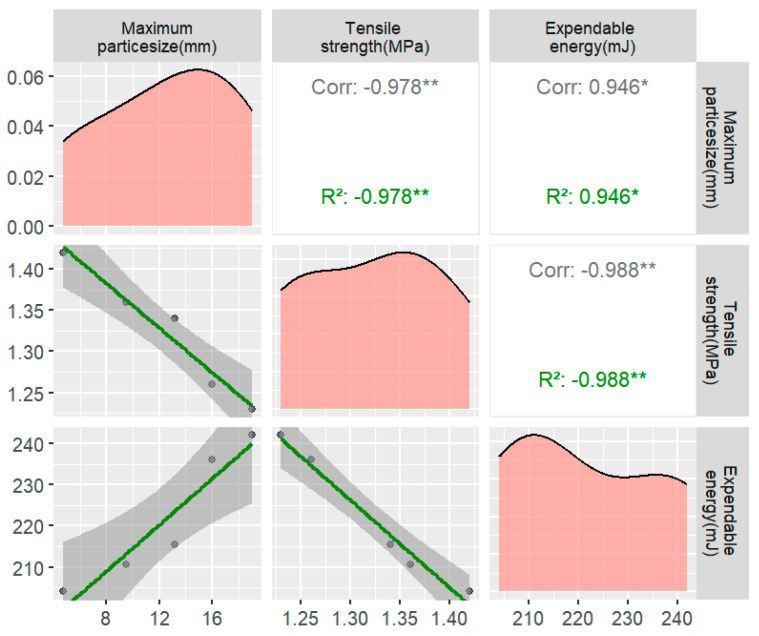
Plot of the tensile strength and energy dissipation for different coarse aggregates with the maximum particle size (* usually indicates that the *p*-value lies between 0.05 and 0.01, indicating statistical significance, but not very high significance. ** probability value of *p* less than 0.01 indicates high statistical significance).

**Figure 11 materials-17-02627-f011:**
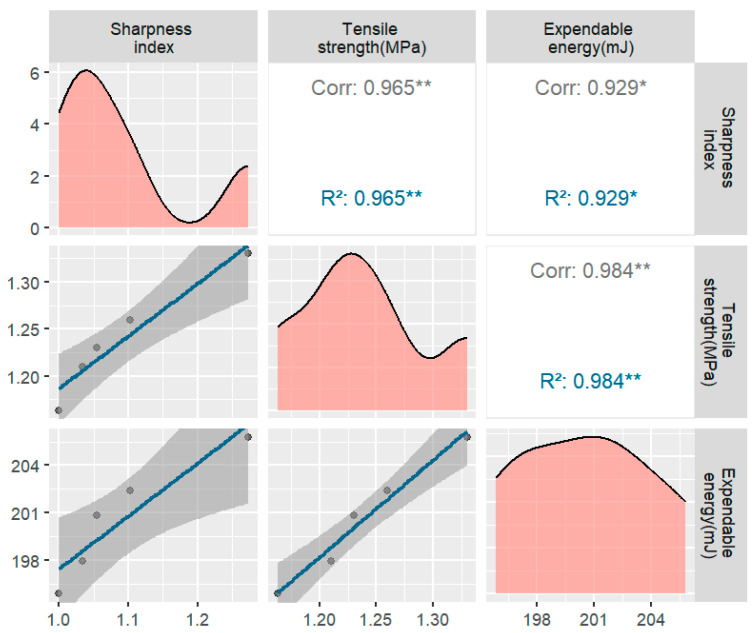
Plot of the tensile strength and energy dissipation for different prismatic indices (* usually indicates that the *p*-value lies between 0.05 and 0.01, indicating statistical significance, but not very high significance. ** probability value of *p* less than 0.01 indicates high statistical significance).

**Table 1 materials-17-02627-t001:** Karamay 70 Grade A asphalt selection criteria table.

Serial Number	Identification of Items		Unit
1	Penetration (25 °C, 100 g, 5 s)		1/10 mm
2	Softening point (ring-and-ball test)		°C
3	Ductility (10 °C, 5 cm/min)		cm
4	Density (20 °C)		g/cm^3^
5	Bitumen film Heating test	Mass changes	%
		Needle penetration ratio	%
		Ductility (10 °C, 5 cm/min)	cm

**Table 2 materials-17-02627-t002:** Coarse aggregate selection criteria table.

Serial Number	Items	Unit	Design Requirements	Test Results
1	Apparent density	g/cm^3^	≥2.6	2.72
2	Adhesion to asphalt	Level	≥4	4
3	Needle-flake particle content	%	≤25	6.97
4	Crushing value	%	≤30	9.56
5	Water absorption	%	≤2	0.43
6	Mud content	%	≤0.5	0
7	Durability (Na_2_ CO_3_ loss of mass in 5 dry and wet cycles)	%	≤12	4.2

Note: The requirement indexes are derived from the requirement indexes for asphalt concrete coarse aggregate in the water conservancy industry standard of the People’s Republic of China, Design Code for Asphalt Concrete Panels and Core Walls of Earth and Stone Dams (DL/T 5411-2009 [37]).

**Table 3 materials-17-02627-t003:** Fine aggregate material selection table.

Serial Number	Sports Event	Unit (of Measure)	Required Indicators	Test Results
1	Apparent density	g/cm^3^	≥2.55	2.71
2	Water absorption	%	≤2	-
3	Water stability rating	classifier: step, level	≥6	6
4	Durability (loss of mass from 5 wet and dry cycles of sodium sulfate)	%	≤15	1.2
5	Stone powder content	%	<5	

**Table 4 materials-17-02627-t004:** Filler selection criteria table.

Serial Number	Sports Event		Unit (of Measure)	Required Indicators	Test Results
1	Apparent density		g/cm^3^	≥2.5	2.715
2	Moisture content		%	≤0.5	---
3	Hydrophilicity		---	≤1.0	0.628
4	Fine particulate degree (angles, temperature etc.)	<0.6 mm	%	100	100
		<0.15 mm		>90	99.7
		<0.075 mm		>85	86.5

Note: The requirement indexes are derived from the requirement indexes for asphalt concrete filler in the water conservancy industry standard of the People’s Republic of China, “Design Code for Asphalt Concrete Panels and Heart Walls of Earth and Stone Dams” (DL/T 5411-2009).

**Table 5 materials-17-02627-t005:** Parameters of the concrete components.

Makings	Aggregate	Mortar (Building)	Asphalt Mortar–Asphalt Mortar	Asphalt Mortar–Coarse Aggregate
Modulus of elasticity (MPa)	60,000	20,000	-	-
Poisson’s ratio	0.2	0.2	-	-
Tensile strength (MPa)	-	-	3.8	1.9
Stiffness (MPa/mm)	-	-	2.0 × 10^7^	1.8 × 10^7^
Fracture energy (N/mm)	-	-	0.09	0.045

**Table 6 materials-17-02627-t006:** Inversion table of the interfacial bond strength parameters.

Storing Time (Days)	Tensile Strength (MPa)	Tensile Modulus (MPa)	Bonding Strength (MPa)
0	1.71	83.24	5.38
10	1.57	116.91	4.85
20	1.07	128.77	3.15
30	0.91	148.57	2.66

**Table 7 materials-17-02627-t007:** Asphalt concrete tensile test results.

Specimen Type	Storing Time (d)	Intensity (g/cm^3^)	Maximum Tensile Strength R_tmax_ (MPa)	Strain at Maximum Tensile Strength ε_max_ (%)	Tensile Modulus (MPa)
No anti-stripping agent	0	2.423	1.711	2.80	83.24
	10	2.417	1.573	3.17	116.91
	20	2.403	1.073	2.55	128.77
	30	2.389	0.914	1.45	148.57
Add SK-A anti-stripping agent	0	2.425	1.723	2.79	85.39
	10	2.421	1.609	2.94	113.79
	20	2.412	1.324	3.25	119.19
	30	2.401	1.294	3.41	146.46

## Data Availability

Data are contained within the article and Appendix A.

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
