# Peer review of "Study on the Effect of Asphalt Static Conditions on the Tensile Properties of Acidic Aggregate Hydraulic Asphalt Concrete"

_materials, 2024, doi:10.3390/ma17112627_

Round 1
Reviewer 1 Report
Comments and Suggestions for Authors
The authors carried out both indoor experiments and numerical simulations to evaluate the tensile properties of acidic aggregate hydraulic asphalt concrete under different static conditions. The manuscript is interesting. However, a major revision is required before publication. Main concerns are as follows:
1. It is necessary to thoroughly revise the abstract section to succinctly convey the research study's significance, the nature of the investigations conducted, the outcomes achieved, and the associated advantages and importance. Please consider that the abstract should be a total of about 200 words maximum.
2. Also the introduction section requires substantial improvement. The authors are strongly encouraged to incorporate additional information and relevant studies to enhance introduction quality (20 bibliographic references are very few). Please consider that the introduction should primarily connect the current state-of-the-art to the goals of the paper. Furthermore, the concluding part needs to be properly formatted. Finally, it seems that the main goals are first described in words and then with a numbered list, redundantly. Please check, revise and improve.
3. Be careful to use the same number of decimal digits for Bonding strength (MPa) in Table 2.
4. In Table 3, the unit of measurement for Standing time (storing time?) is missing. Please check and revise.
5. Try to be as consistent as possible with vocabulary throughout the manuscript in terms of standing or storing time, antiscaling or antispalling agent, and so on.
6. In Figure 7(b) the correct units for storing time should be (d) not (MPa). Please check and revise.
7. In Figures 8, 9, 10, 11, the units of measurement on both x- and y-axes are totally missing. Please check and revise.
8. Discuss the limitations of this work in the discussion and conclusion sections. Identify and address any constraints or shortcomings in the study to provide a balanced perspective on the research.
9. A bibliographic reference is missing at lines 521-522.
10. In the conclusion section, potential future developments are completely missing. Please check and revise.
Author Response
请参阅附件。

Reviewer 2 Report
Comments and Suggestions for Authors
1. In the Introduction, the issue of adhesion between bituminous binder and acidic stone materials should be discussed in more detail, and possible solutions to this problem should be presented.
2. It was worth investigating the use of lime mineral powder as a filler. The use of such material improves the water resistance of asphalt concrete.
3. It is necessary to provide a more detailed description of the newly used anti-scaling agents, including their appearance (color, smell), viscosity, etc.
4. The article should explain how the test results obtained by the proposed method relate to the determination of the water sensitivity of bituminous specimens according to EN 12697-12.
Reviewer 3 Report
Comments and Suggestions for Authors
materials-2937682:
*The title of the article is confusing. Asphalt static conditions? What is that? Make it clearer.
*Reduce the introductory part of the abstract; It is very long. Overall, the rest of the abstract is adequate.
*The keywords are adequate.
*The introduction is very long. Furthermore, make clear the impact of this investigation on the area of knowledge.
*2. Materials and Methods:
- Where is Table S1, S2, S3, S4 ….? These tables have material characteristics and must be in the document.
- Introduce the properties of Anti-scaling agents.
- The authors forget to mention several standards. Review the entire document.
- Justify the parameters used for the materials (table 1).
- Figures 3, 4 and 5 are disproportionate; reduce the size.
*3. Results and Discussion:
The results were well discussed. The article presents very interesting data. However, little comparison with the literature was established. Therefore, it would be important for the authors to compare the findings with the existing literature.
*Conclusions are supported by the data.
*Most references are current and adherent to the topic. However, authors can expand the literature review to compare their findings with results already documented in the literature.
Comments on the Quality of English LanguageModerate editing of English language required.
Round 2
Reviewer 1 Report
Comments and Suggestions for Authors
Comments provided by reviewer have been more or less properly accomplished.
Reviewer 3 Report
Comments and Suggestions for Authors
materials-2937682R1:
The authors performed a good review of the paper.
Comments on the Quality of English LanguageMinor editing of English language required.
